# TeLLMate: Trusted Execution for Large Language Models at the Edge

## Abstract

Large Language Models (LLMs) are being increasingly deployed on edge devices to support real-time inference, preserve user privacy, and enhance energy efficiency. However, on-device deployment creates a lucrative attack surface for adversaries to steal the model. Trusted Execution Environments (TEEs) provide hardware-isolated enclaves to safeguard model parameters. However, existing approaches either demand costly retraining from substantial architectural modifications or incur significant communication overhead by protecting parameters across all layers within the TEE. To address these limitations, we propose TeLLMate, which helps identify the critical layers of LLMs for TEE protection. TeLLMate delivers two key capabilities: (1) a methodology for identifying layers critical to model performance using mathematical analysis. (2) a selective protection algorithm that minimizes the secure memory footprint while defending against retraining attacks. Our experimental results demonstrate that TeLLMate offers strong protection guarantees, resulting in at least a $15\times$ increase in perplexity and close to random-guess accuracy in zero-shot downstream tasks for the attacker's replicated model by including at most 10% of the LLM's parameters.

## 1 Introduction

Large Language Models (LLMs) (Touvron et al., 2023a; Workshop et al., 2023) have demonstrated exceptional performance in a wide range of applications (Noy & Zhang, 2023; Brown et al., 2020). To protect user privacy and eliminate network-transfer latency (e.g., delays from weak or variable mobile signals), LLMs are increasingly run directly on edge devices (Qu et al., 2025). This trend is particularly prominent in privacy-sensitive applications, such as healthcare, finance, wearables, smart home assistants, and automotive systems, where on-device deployment of the LLM model eliminates the need to transmit sensitive user data to the cloud. Despite these benefits, LLM inference on-device also introduces new security vulnerabilities. Unlike centralized cloud servers, edge devices are often physically accessible to users. This makes it possible for adversaries to directly extract model parameters through techniques such as side channel leakage (Batina et al., 2019), illegal memory accesses (Sun et al., 2021), or query-based model extraction (Jagielski et al., 2020) to carry out a model-stealing attack. Once extracted, the stolen model can be redistributed, deployed on unauthorized devices, or exploited for financial gain. Considering the substantial resources and engineering efforts involved in training LLMs (Cottier et al., 2024), the Intellectual Property (IP) of the models represents a valuable asset and must be protected.

One promising approach is to put the model in the Trusted Execution Environment (TEE). TEE is an isolated, secure enclave that protects the data and code inside with respect to confidentiality and integrity (Lee et al., 2020). However, deploying the entire model inside a TEE for protection is impractical due to its limited computing and memory resources. For example, even a small DNN model suffers a roughly $50\times$ increase in latency due to the limited computational capabilities of TEEs (Tramer & Boneh, 2018). To leverage TEEs for model protection, a common strategy is to partition the model. However, existing partitioning methods are ill-suited for LLMs. Techniques developed for smaller DNNs, such as Mirrornet (Liu et al., 2023) and TBNet (Liu et al., 2024), require specialized architectural modifications and costly retraining, making them impractical for large-scale LLMs. More recently, TEESlice demonstrates the potential of utilizing TEEs to protect LLMs by training lightweight, layer-wise modules (i.e., slices) with private data to be placed within the TEE, while retaining the pre-trained backbone LLMs in the REE. However, TEESlice allocates

protected slices uniformly across all layers, resulting in substantial communication overhead between the TEE and REE. As a result, additional weight obfuscation techniques are employed to enhance security by applying reversible transformations to weight matrices before they are stored or executed in untrusted environments. However, transformations increase computational and memory overhead, and attackers may still be able to infer the original weights by analyzing patterns. These challenges raise a fundamental question: *how can we minimize the number of layers placed within the TEE while still ensuring strong resistance against retraining attacks?*

In this work, we propose TeLLMate, a novel algorithmic approach that identifies the most critical layers of an LLM and strategically places them within the TEE, while executing the remaining layers outside the enclave for efficiency. TeLLMate makes the following key contributions:

- **Critical Layer Identification:** Inspired by the well-known phenomenon of outlier activations in LLMs (Dettmers et al., 2022; Sun et al., 2024), we identify *critical outliers* in a few specific layers across various LLMs. These outlier-associated layers (i.e., critical layer) serve as the functional core of the model, and protecting them from adversaries significantly degrades the performance of any stolen replicate model.

- **Selective Protection Algorithm:** Our algorithm pinpoints the *smallest* set of consecutive layers that must run inside the TEE to thwart retraining-based model stealing attacks. By safeguarding only this subset, it slashes the secure-memory footprint and reduces the communication cost between TEE and REE.

- **TeLLMate** achieves strong protection for various LLMs by protecting only an exceptionally small subset of layers (e.g, only 2 out of 32 layers in LLaMA 2-7B). This approach results in at least a $15\times$ increase in perplexity and random-guess zero-shot accuracy for the attacker's surrogate model while including at most 10% of the LLM's parameters.

## 2 BACKGROUND AND RELATED WORK

### 2.1 MODEL PROTECTION IN THE TRUSTED EXECUTION ENVIRONMENT (TEE)

Trusted Execution Environment (TEE) is a hardware enclave that is isolated from the operating system. It provides hardware-enforced protection to ensure the confidentiality and integrity of user code and data. Prominent TEE implementations include Intel SGX (Costan & Devadas, 2016), ARM TrustZone (Alves & Felton, 2004), and RISC-V Keystone (Lee et al., 2020). In this paper, we follow prior work and consider the TEE as a secure enclave on a potentially adversarial host device, ensuring that the data, code, and computations within the TEE are protected. In contrast, the Rich Execution Environment (REE) refers to the normal operating system and application space. It has full access to system resources but lacks the security guarantees (GlobalPlatform, 2018). Existing efforts to protect machine learning models with TEEs can be grouped into two main categories:

**Partition-based methods.** These approaches split the model into two parts and execute the sensitive portion inside the TEE. DarkneTZ (Mo et al., 2020) partitions CNN models by executing the first few sensitive layers inside the TEE while leaving the remaining layers in the REE. However, its protection strategy is largely empirical and lacks a systematic methodology to determine how many layers should be protected. Mirrornet (Liu et al., 2023) and TBNet (Liu et al., 2024) restructure DNNs into dual-branch architectures, then retrain the two-branch model and deploy the lightweight branch inside TEE for model protection. While effective for small DNNs, the high costs of architectural modifications and retraining make these methods impractical for LLMs. More recently, TEESlice (Li et al., 2025) shows the potential to protect LLMs on TEE. It partitions models before training by fixing a public backbone and training only a private lightweight model (i.e., slice) whose weights are stored in the TEE. However, slice models are across all layers of the public backbone, and they require frequent communication between TEE and REE. Also, the vulnerability of protected slice models to model stealing remains unknown, necessitating the use of weight obfuscation for stronger protection. In contrast to prior works, we propose an outlier-aware layer selection method tailored for LLMs, which places only the minimum number of critical layers into the TEE to accommodate its limited resources while remaining strongly resistant to model stealing attacks.

**Weight obfuscation methods.** Another line of work obfuscates model weights so that they can be stored outside the TEE without revealing their true values. ShadowNet (Sun et al., 2023b) obfuscates

model weights by applying linear transformations and outsourcing the heavy computation to untrusted accelerators, with the results later restored inside the TEE. However, such linear transformations are vulnerable to strong attackers, who can monitor memory access patterns, correlate transformation pairs, and ultimately recover the original weights (Li et al., 2025). CoreGuard(Li et al., 2024b) and TransLinkGuard(Li et al., 2024a) row-permute the weight matrices of linear layers, while the TEE applies the corresponding column permutation to the input features. Additionally, they encrypt the input features so that only the TEE can correctly decrypt and align them, thereby preventing the attacker from inferring the permutation order. Similarly, Game of Arrows (Wang et al., 2025) demonstrates that existing weight obfuscation techniques (e.g., scaling, permutation) are insufficient, as an attacker can still recover the model functionality. They propose more complex matrix–vector transformations to strengthen the weight obfuscation. Note that in these schemes, only the activation function is executed inside the TEE, and there are no weight parameters stored within the TEE. These obfuscation techniques are complementary to our method and could be combined to further strengthen security.

## 2.2 Outlier Activations in LLMs

*Outlier activations* (Dettmers et al., 2022; Sun et al., 2024) have been widely observed in Large Language Models (LLMs), referring to the phenomenon where an extremely small subset of activation values (e.g., 1% of the entire activation values) exhibit magnitudes significantly larger than the average across the same layer. These extreme values, while sparse, have a disproportionate influence on model behavior and pose unique challenges for model compression techniques, particularly quantization and pruning. In quantization, which aims to compress the high precision weight/activation (e.g., 16-bit in floating point) to lower precision (e.g, 4-bit in integer), outliers increase the dynamic range, leading to severe rounding or clipping errors that degrade model accuracy. To address this, several outlier-aware quantization methods have been proposed. For example, SmoothQuant (Xiao et al., 2023) rescales the activations and weights to align their magnitude, reducing the quantization error. In the context of pruning, which aims to remove a portion of unimportant weights, outliers complicate magnitude-based importance estimation, as naively pruning them can result in significant performance degradation. To mitigate this, pruning methods such as Wanda (Sun et al., 2023a) incorporate activation statistics to indicate the important weights that need to be preserved. Unlike existing methods that utilize outlier activations to improve LLM compression through pruning or quantization, our work is the first to leverage outlier activations to identify and protect critical layers for secure on-device inference.

## 3 Threat Model

We consider a threat model involving two entities: a defender and an attacker. The defender owns the model deployed on an edge device, while the attacker aims to extract or replicate the model through unauthorized access to the device.

**Attacker's Goal and Capability:** The goal of the attacker is to steal the model, and the success metric is to get functionally equivalent weights that could be used to be deployed on the unauthorized device for financial use. We assume a strong adversary with full access to everything outside the TEE, including the ability to execute inference and observe inputs and outputs at the TEE boundary. The attacker is assumed to have full knowledge of the model architecture protected by the TEE. Although this is not practical in real-world scenarios, it allows for rigorous security analysis. With access to part of the dataset, attackers attempt to reconstruct the protected components of the model through model retraining techniques, such as LLM-Streamline (Chen et al., 2024) and Knockoff Nets (Orekondy et al., 2019).

**Defender's Goal and Capability:** Following prior work (Zhang et al., 2024a) (Zhang et al., 2024b) (Sun et al., 2023b), we assume a hardware-enforced TEE that protects confidentiality and integrity for data inside it, even though the edge device itself is controlled by a malicious adversary. The defender's objective is to ensure that only authorized users, verified by the device's TEE, can access the model. Due to the high cost of retraining LLMs, the defender cannot modify the model architecture or retrain it. Instead, they aim to prevent attackers from reconstructing or transplanting the model.

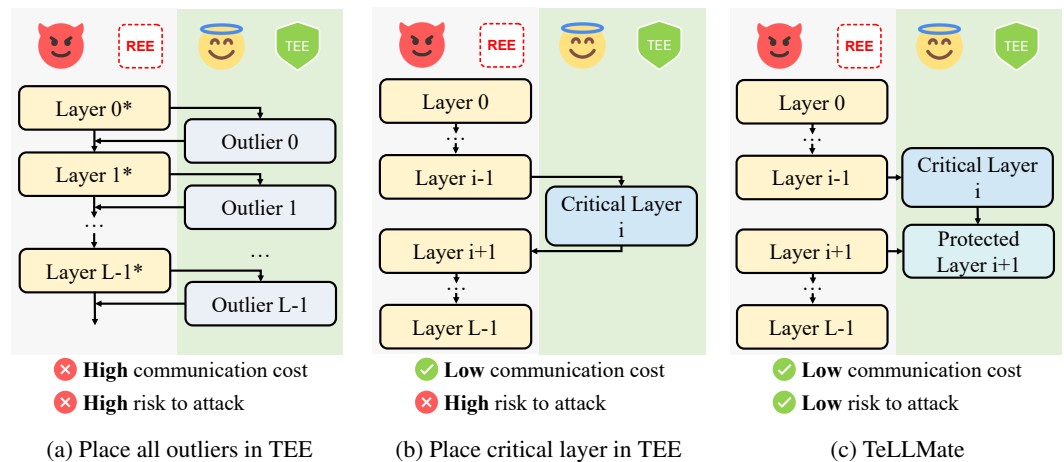

(a) Place all outliers in TEE     (b) Place critical layer in TEE     (c) TeLLMate

Figure 1: Model partitioning strategies for LLM protection. * indicates the decoder layer after removing outlier-associated weights.

# 4 TELLMATE

## 4.1 *Critical outlier:* LLM COLLAPSE BY REMOVING 0.02% WEIGHTS

Intuitively, based on the outlier activation phenomenon, the weights associated with these outliers are particularly critical from a security perspective and thus warrant stronger protection than other weights. A straightforward approach to protect LLMs is to place all outlier-associated weights in Trusted Execution Environment (TEE), while assigning the remaining weights to the Rich Execution Environment (REE) as shown in Figure 1a. However, such a design poses a high risk of information leakage and incurs substantial communication overhead between the TEE and REE, as each layer must be partially computed across both environments, leading to frequent intermediate activation exchange.

To address this issue, we begin by exploring the characteristics of outlier-associated weights in a layer-wise manner. In particular, we conduct experiments on various LLMs to examine whether outliers in different layers contribute equally. To identify the outlier-associated weights in each layer, we adopt Wanda (Sun et al., 2023a), an activation-aware pruning method that leverages the product of weight magnitude and activation as an importance score to remove weights. As shown in Figure 2, it's particularly intriguing to observe that:

*Removal of outlier-associated weights for a few specific layers induces significant performance degradation or even model collapse, while similar removal from other layers has less effect.*

We name these outliers in these specific layers with significant performance degradation as **critical outliers**.

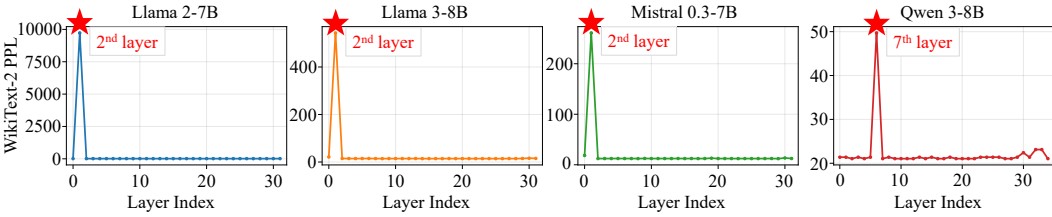

Figure 2: We evaluate the WikiText-2 PPL on four different LLMs by removing the same amount of the highest outliers from each layer. The parameter size of these outliers only accounts for 0.02% of the whole model. The actually important outliers in specific layers (marked as critical layers) are defined as critical outliers.

Table 1: Perplexity of different attack settings on LLaMA 2-7B, with lower perplexity (i.e., PPL) values indicating better performance.

| Critical Component | Surrogate Model | Protected Params Ratio | PPL (Wiki2) | PPL (PTB) |
|---|---|---|---|---|
| Baseline | No attack | 0.00% | 12.19 | 47.21 |
| Critical Outliers | Outliers | 0.02% | 12.72 | 48.30 |
| Critical MLP | MLP | 2.09% | 12.49 | 86.29 |
| Critical Layer | Decoder Layer | 3.13% | **304577.06** | **287170.95** |
| Critical Layer | **MLP** | 3.13% | 26.36 | 153.16 |

## 4.2 LLM PROTECTION BY SECURING THE CRITICAL COMPONENTS IN TEE

The defined *critical outlier* underscores its importance in maintaining LLM performance, making it a prime candidate for protection. Building on this insight, we can only place the critical outliers into the TEE while the remaining weights are executed in the REE. By doing so, it ensures that, without access to the TEE-protected critical layer, the model's performance degrades significantly. Furthermore, as the TEE handles only outliers that occupy $\sim 0.02\%$ weights of the model, the overall computation and memory cost remains extremely low, aligning well with the limited resources of the TEE.

The critical outlier is partitioned in the TEE, while the remaining layers are executed in the REE. By doing so, it ensures that, without access to the TEE-protected critical layer, the model's performance degrades significantly. Furthermore, as the TEE handles only one layer, the overall computation and memory cost remains low, aligning well with the limited resources of the TEE.

However, as we mentioned in Section 3, since the TEE-projected components have to communicate with the rest of the model running in the REE, they expose a boundary through which an attacker can observe the input and output activations of the projected components. This exposure enables a model stealing attack - *retraining attack*, where the attacker exploits the observed intermediate states to reconstruct or substitute the functionality of the TEE-projected components by training a surrogate model.

Therefore, it is critical to evaluate the resistance of the TEE-projected components to such retraining attacks. To achieve that, we perform an empirical analysis to study the resistance of the retraining attack for different types of critical components. In particular, we conduct experiments on the LLaMA 2-7B model by gradually increasing the number of parameters placed in the TEE and evaluating the attacker's ability to recover the performance of the model (i.e., PPL). As shown in the Table 1, we evaluate

Figure 3: *Model retraining attack:* The attacker freezes layers outside the protection set and trains a surrogate layer based on the inputs and outputs of the critical component via an MSE loss, thereby bypassing the protected layers.

three critical components in the TEE for retraining attacks beyond critical outliers: (1) **Critical MLP**, which secures the MLP block containing the critical outlier-associated weights; (2) **Critical Layer**, which extends the Critical MLP to further place the attention block in the same decoder layer. Note that, in the standard architecture of LLMs, each decoder layer consists of an attention block followed by an MLP block; and (3) **Critical Layer + Following Layer**, which secures the Critical Layer with one consecutive decoder layer. We summarize our main findings as below:

*1) Securing the critical outliers is not resistant to retraining attacks.* When a retraining attack uses the same architecture, the model with only critical outliers protected shows similar performance to the baseline, indicating limited resistance to surrogate reconstruction.

*2) More weights in TEE enhance resistance to retraining attack.* Compared to Critical Outliers and Critical MLP, Critical Layer has a significantly large PPL on both evaluation datasets, indicating the model collapse that can not be attacked.

*3) MLP surrogate is a stronger retraining attack configuration.* Motivated by recent work LLM-Streamline (Chen et al., 2024), which demonstrates the feasibility of using MLPs to approximate

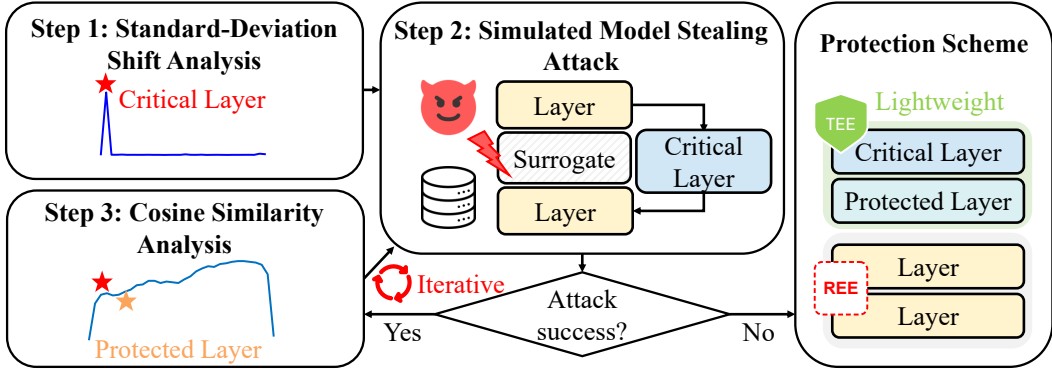

Figure 4: Our workflow of TeLLMate consists of three steps: (1) selecting the initial layer based on standard deviation shift, (2) evaluating its resistance through a simulated model stealing attack, and (3) iteratively selecting additional layers using layer-wise cosine similarity.

decoder layers for model compression, we modify the attacker's surrogate architecture from a decoder layer to an MLP. This change results in modest degradation in model performance, indicating that the attack remains partially successful as shown in the Figure 1b. We hypothesize that two factors explain why the **Critical Layer** surrogate underperforms compared to the **Critical MLP** surrogate. First, retraining a surrogate from scratch is substantially more difficult than model compression via knowledge distillation, where the student model is initialized from the teacher's weights. Here, the surrogate has no access to the protected weights and must be trained from random initialization. Second, the decoder surrogate contains a larger parameter space and self-attention submodules that are harder to optimize; in particular, randomly initialized Q/K/V weights hinder convergence. By comparison, an MLP surrogate has fewer parameters and only position-wise operations, enabling more stable training and making it more effective at approximating the protected layers. Therefore, based on the analysis of the attacking result, we adopt this surrogate configuration in our retraining-based attack targeting the protected decoder layer.

Overall, our findings indicate that to achieve strong resistance against retraining attacks, it is necessary to include additional layers within the TEE beyond the initially selected critical components. However, this enhanced protection comes at the cost of increased latency and is constrained by the limited secure compute resources available inside the TEE (Tramer & Boneh, 2018). Furthermore, the optimal number of layers required for effective protection is unknown and may vary across LLMs depending on their architecture and scale.

## 4.3 ITERATIVE AUTO LAYER SELECTION FOR TEE PROTECTION

To address this issue, we propose a method that automatically selects and places a minimal number of layers into the TEE while ensuring strong resistance against retraining attacks, as shown in Figure 1c. Specifically, the proposed method includes three stages as illustrated in Figure 4, each answering one key question: (1) How can we identify the initial layer to be placed in the TEE? (2) How to quantitatively assess the protective capacity to select more layers against retraining attacks? (3) Which additional layers should be incorporated into TEE to enhance resistance while keeping the overall parameter size minimal?

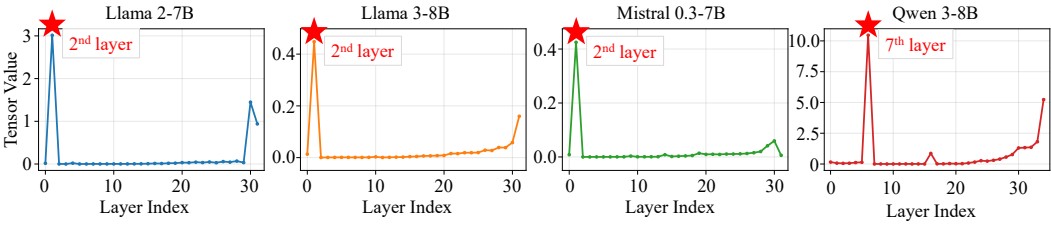

Figure 5: Layer-wise visualization of the standard deviation shift metric in different LLMs.

① **Selecting initial layer via standard deviation shift.** We begin by selecting the *critical layer* to place in the TEE. In the Section 4.1, we identify critical layers using a pruning-based metric, but this approach is computationally intensive and relies on access to the downstream evaluation dataset, making it less practical. To address this, we develop a new metric to identify critical layer by measuring the standard deviation shift in activations between the input and output of each layer:

$$\Delta\sigma_\ell = |\sigma(\mathbf{h}_\ell) - \sigma(\mathbf{h}_{\ell-1})| \tag{1}$$

Where a large $\Delta\sigma_\ell$ suggests that the layer significantly skews the distribution attribute. Empirically, we find that layers that hold spikes in the standard deviation shift are highly likely to be the critical layer. For each decoder layer $\ell$, we compute the standard deviation shift in activations between the input and output of each layer. The layer exhibiting the largest shift is treated as the critical layer, as shown in Figure 5, and it will be necessary to be included by TEE protection.

Although outlier activations appear throughout the model, their impact differs across layers as shown in Figure 2. We argue that the standard deviation shift, which measures the change in activation variability between a layer's input and output, provides a concrete signal of each layer's influence. Since this variability is largely caused by outliers, a large shift suggests that the layer plays a significant role in amplifying them, making it a strong indicator of the outlier contribution across layers.

② **Evaluating resistance via simulated retraining attack.** Once the selected layer is placed in the TEE, TeLLMate evaluates whether it provides effective resistance against retraining attacks. To do this, we simulate a realistic threat model in which an adversary attempts to reconstruct the protected layer using a partial dataset. We use an MLP as the surrogate model for the retraining attack, which represents a more challenging scenario than replicating the original decoding layer architecture, as shown in Table 1. To this end, if the trained model can not achieve the predefined performance, the protection is considered unresistant. In particular, we consider 300 of the Wiki-2 PPL score as the pre-defined target performance. Note that, if the model has a larger than 300 PPL, which is $\sim 20\times$ than the baseline PPL, it's safe to be considered to have collapsed, and we will select this configuration as the optimal protection scheme. Otherwise, we will further add one more layer in the TEE iteratively until the collapse condition is met.

③ **Iteratively selecting additional layers via layer-wise cosine similarity.** To strengthen resistance against model retraining attacks, we expand one additional layer into the TEE for each iteration step. To minimize the number of layers that are placed in TEE and ensure strong resistance, two key criteria must be satisfied: (1) the selected layer should offer greater resistance to retraining attack than others, and (2) it should be directly connected to the existing protected layers to avoid exposing intermediate activations in TEE that can be exploited for retraining attack.

To achieve that, we first draw on insights from previous works on layer sensitivity and pruning robustness (Frankle & Carbin, 2018; Gromov et al., 2024), which suggest that weights or layers difficult to prune, those whose removal leads to significant performance degradation, are also harder to approximate. Such that, the proposed standard deviation shift is a potential metric to select more layers. However, as shown in the Figure 5, although it can clearly indicate a few critical layers, the rest of the layers often have similar values, making them difficult to distinguish. To address this limitation, we draw inspiration from recent layer pruning works such as ShortGPT (Men et al., 2024) and LLM-Streamline (Chen et al., 2024), which show that layers with high cosine similarity between input and output activations tend to contribute less to model performance and are thus more suitable for pruning. Building on this, TeLLMate examines layers adjacent to the critical layer or current protected set $\mathcal{P}$. For each candidate, we compute cosine similarity between the input and output activations, which can be formulated as:

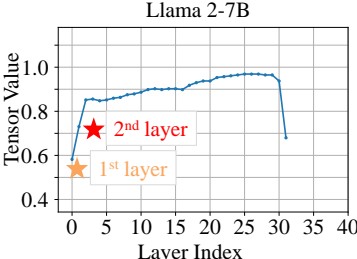

Figure 6: Cosine similarity guided selection.

$$\cos\_sim(\mathbf{h}_{\ell-1}, \mathbf{h}_\ell) = \frac{\mathbf{h}_{\ell-1}^\top \mathbf{h}_\ell}{\|\mathbf{h}_{\ell-1}\|_2 \|\mathbf{h}_\ell\|_2}. \tag{2}$$

After that, TeLLMate appends the candidate with lower cosine similarity to the protection set. This creates a new configuration of the protection scheme and TeLLMate goes to Step 2 for simulation.

Table 2: Overall attack performance evaluation on text generation perplexity and average CMQA 0-shot accuracy of various LLMs under different protection settings. The better the protection, the lower the performance.

| Model | Method | Selected Layer index | PPL (Wiki2) ↑ | PPL (PTB) ↑ | CMQA ↓ | Params Ratio | FLOPs Ratio |
|---|---|---|---|---|---|---|---|
| LLaMA 2-7B | Baseline | – | 12.19 | 47.21 | 66.79 | 0.00% | 0.00% |
| | TeLLMate-C | [1] | 26.36 | 153.16 | 53.27 | 3.13% | 3.13% |
| | TeLLMate-F | [0,1] | 4710.54 | 3559.06 | 36.19 | 6.25% | 6.25% |
| LLaMA 2-13B | Baseline | – | 10.98 | 56.15 | 69.27 | 0.00% | 0.00% |
| | TeLLMate-C | [3] | 12.22 | 59.49 | 68.82 | 2.50% | 2.50% |
| | TeLLMate-F | [0,1,2,3] | 4079.78 | 5108.48 | 35.03 | 10.00% | 10.00% |
| LLaMA 3-8B | Baseline | – | 14.14 | 27.96 | 70.17 | 0.00% | 0.00% |
| | TeLLMate-C | [1] | 20.37 | 36.95 | 66.32 | 3.13% | 3.13% |
| | TeLLMate-F | [0,1] | 1007.44 | 1284.55 | 35.88 | 6.25% | 6.25% |
| Mistral 0.3-7B | Baseline | – | 15.14 | 31.12 | 70.39 | 0.00% | 0.00% |
| | TeLLMate-C | [1] | 262.12 | 438.91 | 35.44 | 3.13% | 3.13% |
| | TeLLMate-F | [0,1] | 217695.49 | 386990.95 | 38.99 | 6.25% | 6.25% |
| Qwen 3-8B | Baseline | – | 21.13 | 42.36 | 69.67 | 0.00% | 0.00% |
| | TeLLMate-C | [6] | 102.00 | 184.70 | 50.43 | 2.78% | 2.78% |
| | TeLLMate-F | [6,7] | 324.16 | 753.70 | 40.13 | 5.56% | 5.56% |

The loop iterates until either (i) the attack fails or (ii) a user-defined protection budget $L_{\max}$ is reached. The final protection set thus represents the minimal collection of layers that must be protected within the TEE against retraining-based model-stealing attacks, considering resource constraints.

## 5 EXPERIMENT

### 5.1 EXPERIMENT SETUP

We evaluate the effectiveness of TeLLMate by assessing the performance of surrogate models (i.e. PPL) trained by the attacker, as discussed in Section 4.2. Specifically, we conduct experiments using five open-source LLMs: LLaMA 2-7B, LLaMA 2-13B (Touvron et al., 2023b), LLaMA 3-8B (Grattafiori et al., 2024), Mistral 0.3-7B (Jiang et al., 2023), Qwen 3-8B (Yang et al., 2025). Following a similar experimental setup as prior work (Chen et al., 2024), we use 1% of the SlimPajama-6B dataset (DKYoon/SlimPajama-6B (Soboleva et al., 2023)) to train the attacker's surrogate model. This corresponds to approximately 30,000 samples, each containing 2048 tokens, totaling around 0.06 billion tokens. Detailed training hyperparameters are provided in the Section A.2.

We compare model protection efficacy under three configurations: (1) **Baseline**, which is the original dense model; (2) **TeLLMate-C**, which protects only the critical layer identified through standard deviation shift analysis in Section 4.2; and (3) **TeLLMate-F**, which applies the full TeLLMate to generate an optimal protection scheme based on the iterative selection workflow described in Section 4.3. We evaluate model functionality using three metrics: perplexity (PPL) on WikiText-2 (Merity et al., 2016) (Wiki2), perplexity on Penn Treebank (Marcus et al., 1993) (PTB), and zero-shot accuracy on the CommonsenseQA (CMQA) benchmark suite including BoolQ (Clark et al., 2019), PIQA (Bisk et al., 2020), HellaSwag (Zellers et al., 2019), WinoGrande (Sakaguchi et al., 2019), ARC-Easy (Clark et al., 2018), ARC-Challenge (Clark et al., 2018), and OpenbookQA (Mihaylov et al., 2018), applied through the EleutherAI LM Harness pipeline (Gao et al., 2024).

### 5.2 EXPERIMENTAL RESULTS

The results in Table 2 highlight the strong effectiveness of our framework. Across all models, applying our full protection TeLLMate-F results in a substantial increase in perplexity, demonstrating strong resistance to model retraining attacks. For example, the perplexity of LLaMA 3-8B on WikiText-2 rises from 14.14 to 1007.44 (71 × increase). Notably, for Mistral 0.3-7B, the perplexity increases by about 14k times, indicating that our method effectively collapses the model performance. Moreover, we find that this strong protection can be achieved with minimal iterations of TeLLMate. In most cases, securing just two decoder layers suffices to induce substantial degradation. Notably, the selected layers tend to appear in the early stages of the network, aligning with our findings in Section 4.1 and prior work on layer sensitivity in transformer models (Ma et al., 2023). Moreover, we also observe that the selected protected layers differ across model architectures. For instance, in Qwen 3-8B the

protected layers are [6,7], while they are [0,1] in LLaMA 2-7B and Mistral 0.3-8B models. For LLaMA 2-13B, four layers are selected into the protected set $\mathcal{P}$, corresponding to 10% of the model's parameters. This is a reasonable proportion given the model's larger size and deeper architecture.

Beyond perplexity, we evaluate zero-shot generalization using CMQA, a composite benchmark of seven common-sense reasoning tasks. As shown in Table 2. LLaMA 2-7B's overall CMQA accuracy drops from 66.79% to 36.19% under full protection, demonstrating that our method significantly disrupts model reasoning ability and usability in downstream tasks. Despite its strong effectiveness, our full method maintains low overhead. Across all models tested close to the scale of 7-8B parameters, the parameter protection ratio remains under 6.25%, making our approach lightweight and suitable for deployment in resource-constrained secure environments such as TEE. Detailed accuracy on specific tasks is provided in the Section A.6.

Table 3: Perplexity results on Wiki2 and PTB for LLaMA2-13B under different 4-layer protection configurations.

| Selected Layer index | PPL (Wiki2) |
|---|---|
| Dense | 10.98 |
| TeLLMate-F [0,1,2,3] | 4079.78 |
| Random [3, 4, 5, 6] | 16.82 |
| Random [8,9,10,11] | 12.30 |
| Random [10,11,12,13] | 12.23 |
| Random [15,16,17,18] | 12.25 |

In comparison, the TeLLMate-C leads to mild performance degradation. For example, in LLaMA 2-13B, WikiText-2 perplexity increases from 10.98 to 12.22. This moderate degradation highlights the limitations of single critical layer protection, as the attackers can get a partially functional surrogate model under this configuration. These observations align with our analysis in Section 4.2.

## 5.3 EFFECTIVENESS OF AUTO LAYER SELECTION

To further validate the proposed layer selection algorithm, we compared the protection set TeLLMate-F against four randomly selected four-layer configurations on LLaMA2-13B. As shown in Table 3, none of the random selections achieved performance comparable to that of the protection set identified by TeLLMate, highlighting the benefit of our targeted selection strategy.

In addition, we conduct an ablation study to investigate the impact of cosine similarity guidance. As shown in Table 4, we compare three protection configurations: (1) protecting only the critical layer (TeLLMate-C), (2) adding an adjacent layer not selected by our method, and (3) adding an additional layer selected based on cosine similarity. Results show that cosine-guided selection leads to significantly stronger protection. For example, in the LLaMA 3-8B model on WikiText-2, adding a non-guided adjacent layer increases perplexity slightly from 20.37 to 25.39. In contrast, incorporating the cosine-selected layer

Table 4: Impact of Cosine Similarity Guidance for The Protected Layer Section on Model Perplexity.

| Model | Strategy | PPL (Wiki2) | PPL (PTB) |
|---|---|---|---|
| | TeLLMate-C | 26.36 | 153.16 |
| LLaMA 2-7b | w/o cosine | 346.04 | 1216.56 |
| | w/ cosine | **4710.54** | **3559.06** |
| | TeLLMate-C | 20.37 | 36.95 |
| LLaMA 3-8b | w/o cosine | 25.39 | 45.25 |
| | w/ cosine | **1007.44** | **1284.55** |
| | TeLLMate-C | 262.12 | 438.91 |
| Mistral 0.3-7b | w/o cosine | 111.34 | 262.79 |
| | w/ cosine | **194064.39** | **291104.64** |

leads to a dramatic increase in perplexity to 1007.44, about 50×. The result highlights that naive inclusion of nearby layers without our method may offer only marginal security benefits, while our cosine similarity-based approach enables a targeted and data-efficient protection scheme that maximizes defense strength under strict memory constraints.

## 6 CONCLUSION

We propose TeLLMate, a secure and efficient framework that leverages TEE to protect the LLM against model retraining attacks. TeLLMate identifies the critical layer for model performance and employs an iterative algorithm to progressively expand the protected set. The generated optimal protection scheme balances the trade-off between latency and security. Experimental results show that our method achieves strong protection guarantees, increasing the attacker's model perplexity by at least 15× and random-guess zero-shot accuracy while requiring only 10% of the model's parameters to be enclosed within the TEE.

**Reproducibility Statement.** We provide an anonymous implementation of our method, including scripts for critical layer identification, protection scheme generation, retraining attack, and evaluation scripts, in the Supplementary Material of the submission. Additional details, such as hyperparameters and experimental settings, are described in the main text and appendix to ensure full reproducibility.

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

# A  APPENDIX

## A.1  LLM USAGE

In accordance with the ICLR 2026 policy on large language model (LLM) usage, we disclose that LLM-based tools (e.g., ChatGPT) were used solely to aid in polishing the writing and improving the clarity of exposition. They were not used for research ideation, experimental design, data analysis, or other substantive contributions. All scientific content, results, and conclusions are the responsibility of the authors.

## A.2  DETAILED EXPERIMENTAL SETUP

To conduct the retraining attack, a 1% subset of SlimPajama-6B is used, consisting of 30,000 training samples (each with 2048 tokens, approximately 0.06B tokens) and 3,000 evaluation samples of the same length. Training is conducted for 20 epochs with a batch size of 32. The learning rate is set to 1e-3, with a minimum learning rate of 2.5e-5 and a maximum of 1e-3. Weight decay is configured to 1e-4. We adopt a cosine scheduler to perform learning ratio scheduling. We have a 3% warmup step in the first epochs. All model parameters are set to bfloat16 precision by default if no other notification exists. All experiments are conducted on a cloud computing server with an AMD EPYC 9554 CPU, 318.6 GB of memory, 400GB NVME SSD, and one Nvidia H100 80GB GPU.

## A.3  ALGORITHM BLOCK

---

**Algorithm 1** Iterative Auto Layer Selection for TEE Protection

---

**Input:** Pre-trained LLM with $L$ decoder layers $\{\ell_1, \ldots, \ell_L\}$;
Attack simulation routine RETRAINATTACK$(\cdot)$ returning perplexity;
Target perplexity PPL$_{target}$ (default = 1000).
**Output:** Minimal protection set $\mathcal{P}$ to be enclosed in the TEE.

---

1  /\*Stage 1: Select initial layer via std-dev shift\*/
2  **for** $\ell \leftarrow 1$ **to** $L$ **do**
3    Compute $\Delta\sigma_\ell$ with Eq. (1);
4    $\ell^\star \leftarrow \max\limits_{\ell} \Delta\sigma_\ell$
5    $\mathcal{P} \leftarrow \{\ell^\star\}$

6  /\*Stage 2: Evaluate resistance via retraining attack\*/
7  **Function** Evaluate($\mathcal{P}$):
8    ppl $\leftarrow$ RETRAINATTACK($\mathcal{P}$)        // implement retraining attack
9    **return** ppl
10 **if** $ppl \geq PPL_{target}$ **then**             // attack collapsed
11   **return** $\mathcal{P}$

12 /\*Stage 3: Iteratively enlarge $\mathcal{P}$ via cosine similarity\*/
13 **while** $ppl < PPL_{target}$ **do**
14   $\mathcal{C} \leftarrow$ *adjacent, unprotected* layers of $\mathcal{P}$
15   **foreach** $c \in \mathcal{C}$ **do**
16     Compute $\cos\_sim_c$ with Eq. (2);
17   $c^\star \leftarrow \min\limits_{c \in \mathcal{C}} \cos\_sim_c$
18   $\mathcal{P} \leftarrow \mathcal{P} \cup \{(c^\star)\}$
19   Evaluate($\mathcal{P}$)
20 **return** $\mathcal{P}$

---

## A.4 TeLLMate on Quantized LLMs

Edge devices typically support low-bit arithmetic (e.g., INT4/INT8), making quantization essential for deploying LLMs under resource constraints. To this end, we evaluate whether TeLLMate remains effective when applied to a quantized model on an edge device. Here, we use the AWQ (Lin et al., 2024) framework as a representative quantization method.

We apply TeLLMate to **LLaMA 2-7B AWQ**, a 4-bit quantized model, and compare its performance under different scenarios.

- ✗ indicates the attacker trains the surrogate model using **bf16 precision**, matching the original precision of LLaMA 2-7B without quantization, while ignoring computational overhead.

- ✓ denotes that the attacker also adopts **int4 quantization-aware training (QAT)**, aligning with the precision of the deployed model.

- ↻ represents a setting where the quantization scales are **updated during training** instead of being fixed.

From Table 5, we observe a significant perplexity increase when the attacker is constrained to int4 precision with QAT (✓), compared to using higher-precision (bf16) training (✗). This is attributed to activation noise and approximation errors introduced by quantization that degrade the attacker's ability to recover the protected model's performance. Also, enabling scale updates (↻) provides additional robustness in some settings, likely due to better alignment with the quantization distribution during surrogate training. These results demonstrate that TeLLMate retains strong protective effectiveness under realistic deployment constraints, showing compatibility with low-bit quantized inference and quantization-friendly frameworks.

Table 5: Evaluation of TeLLMate on quantized models. We report performance (perplexity on WikiText-2 and PTB, plus CMQA 0-shot accuracy) for **LLaMA 2-7B AWQ** under various TeLLMate settings. Higher perplexity (↑) and lower accuracy (↓) imply stronger protection.
Legend: "✗" = no QAT (bf16 surrogate), "✓" = QAT with fixed scale, "↻" = QAT with scale update.

| Method | PPL (Wiki2) ↑ | PPL (PTB) ↑ | CMQA 0-shot ↓ |
|---|---|---|---|
| Baseline | 12.19 | 47.21 | 66.79 |
| Baseline (AWQ) | 12.57 | 48.94 | 66.36 |
| TeLLMate-C | 26.36 | 153.16 | 53.27 |
| TeLLMate-C (✗) | 36.37 | 209.30 | 53.41 |
| TeLLMate-C (✓) | 173.51 | 644.68 | 37.78 |
| TeLLMate-C (✓, ↻) | 46.70 | 209.30 | 47.79 |
| TeLLMate-F | 4710.54 | 3559.06 | 36.19 |
| TeLLMate-F (✗) | 5928.34 | 5231.74 | 35.32 |
| TeLLMate-F (✓) | 15138.55 | 20691.95 | 35.12 |
| TeLLMate-F (✓, ↻) | 6717.69 | 7150.95 | 35.29 |

## A.5 Computational Cost Analysis

Regarding the computational cost of our algorithm, we separate the whole process into two parts: protection scheme generation and adversarial attack simulation.

The result of the protection scheme generation is provided in Table 6. This process involves an offline, one-time search for the protected layers, performed once per model, and therefore has no impact on runtime latency or memory during real-time inference.

Furthermore, for adversarial attack simulation, we compared its training time and memory usage against a baseline that retrains the same surrogate model while freezing the entire LLM, similar to standard partial-layer fine-tuning (denoted as "Full LLM"). As shown in the Table 7, our training strategy significantly reduces both training time and memory consumption.

Table 6: Time and memory cost comparison for generating a protection scheme on various target model scales.

| Model | Protection Scheme Time | Protection Scheme Memory |
|---|---|---|
| LLaMA2-7B | 4 min 45 s | 26668 MiB |
| LLaMA2-13B | 9 min 10 s | 42040 MiB |

Table 7: Training attack time and memory cost comparison on various target model scales.

| Model | Method | Loaded Layers | Batch size | Total Training Attack Time | Training Attack Memory |
|---|---|---|---|---|---|
| LLaMA2-7B | Full LLM | 32 | 2 | 3280 min | 45432 MiB |
| | Ours | 2 | 32 | 201 min | 39904 MiB |
| LLaMA2-13B | Full LLM | 40 | 2 | 8000 min | 80678 MiB |
| | Ours | 4 | 32 | 880 min | 80868 MiB |

## A.6 DETAILED ZERO-SHOT ACCURACY OF TELLMATE

We evaluate zero-shot generalization using CMQA, a composite benchmark of seven commonsense reasoning tasks. As shown in Table 8. LLaMA 2-7B's overall CMQA accuracy drops from 66.79% to 36.19% under full protection. On specific tasks like ARC-e, accuracy falls to near-random levels from 76.30% to 26.22%, demonstrating that our method significantly disrupts model reasoning ability and usability in downstream tasks. It is also important to note that BoolQ, PIQA, and ARC-e are binary classification tasks, whereas the remaining tasks involve four classes. Among them, BoolQ has a label imbalance, with approximately 60% positive examples. While TeLLMate-F shows slightly higher accuracy than TeLLMate-C on certain tasks, both models actually have effectively collapsed, as their predictions are no better than random guessing.

Table 8: Detailed accuracy of various LLMs under different attack settings on commonsense QA benchmarks. The better the protection, the lower the performance.

| Model | Method | BoolQ ↓ | PIQA ↓ | Hella. ↓ | Winogr. ↓ | ARC-e ↓ | ARC-c ↓ | OBQA ↓ | Average ↓ |
|---|---|---|---|---|---|---|---|---|---|
| LLaMA 2-7B | Baseline | 77.71 | 78.07 | 76.00 | 68.98 | 76.30 | 46.25 | 44.20 | 66.79 |
| | TeLLMate-C | 70.67 | 67.79 | 54.96 | 57.93 | 58.12 | 33.19 | 30.20 | 53.27 |
| | TeLLMate-F | 40.03 | 53.48 | 30.15 | 50.12 | 26.22 | 27.30 | 26.00 | 36.19 |
| LLaMA 2-13B | Baseline | 80.55 | 79.11 | 79.39 | 72.22 | 79.46 | 48.98 | 45.20 | 69.27 |
| | TeLLMate-C | 80.06 | 78.29 | 78.08 | 72.85 | 78.28 | 49.40 | 44.80 | 68.82 |
| | TeLLMate-F | 37.83 | 52.88 | 25.98 | 47.67 | 25.88 | 28.75 | 26.20 | 35.03 |
| LLaMA 3-8B | Baseline | 81.28 | 79.65 | 79.13 | 72.61 | 80.09 | 53.41 | 45.00 | 70.17 |
| | TeLLMate-C | 79.76 | 76.93 | 76.34 | 67.48 | 75.72 | 47.44 | 40.60 | 66.32 |
| | TeLLMate-F | 40.37 | 56.20 | 28.03 | 49.41 | 29.71 | 21.25 | 26.20 | 35.88 |
| Mistral 0.3-7B | Baseline | 82.08 | 80.25 | 80.44 | 73.88 | 79.63 | 52.22 | 44.20 | 70.39 |
| | TeLLMate-C | 38.38 | 53.97 | 29.42 | 49.41 | 28.79 | 21.93 | 26.20 | 35.44 |
| | TeLLMate-F | 61.87 | 52.99 | 25.97 | 50.59 | 25.17 | 28.92 | 27.40 | 38.99 |
| Qwen 3-8B | Baseline | 86.61 | 76.93 | 74.96 | 67.72 | 83.63 | 56.48 | 41.40 | 69.67 |
| | TeLLMate-C | 72.11 | 63.55 | 53.60 | 54.85 | 49.41 | 29.69 | 29.80 | 50.43 |
| | TeLLMate-F | 52.57 | 58.43 | 33.10 | 50.28 | 35.90 | 24.23 | 26.40 | 40.13 |

## A.7 ROBUSTNESS AGAINST PUBLIC-WEIGHT INITIALIZATION

Table 9: Comparison of perplexity scores across Wiki2, PTB, and CMQA under different initialization methods.

| Method | PPL (Wiki2)↑ | PPL (PTB)↑ | CMQA↓ |
|---|---|---|---|
| Dense | 10.98 | 56.15 | 66.79 |
| Initialized randomly (TeLLMate-F) | 4710.54 | 3559.06 | 36.19 |
| Initialized from corresponding public weight | 4757.53 | 3632.19 | 37.63 |

To evaluate whether access to public pretrained weights strengthens the model stealing attack, we conducted a new experimental study on the LLaMA2-7B model. Specifically, we initialize the surrogate MLP not from random weights, but from the corresponding MLP block in a publicly available LLaMA2-7B model that has been fine-tuned on GSM8K. This simulates a strong adversary who can leverage weights closely aligned with the victim's parameters. As shown in Table 9, even under this enhanced attack model, our protection scheme remains robust, with perplexity scores nearly identical to those obtained with random initialization.

