# OpenReview forum: "TeLLMate: Trusted Execution for Large Language Models at the Edge"
_ICLR.cc/2026/Conference — Submitted to ICLR 2026_

### Official Review · Reviewer_t5h2 · 2025-10-18

**Soundness:** 2
**Presentation:** 3
**Contribution:** 3
**Rating:** 4
**Confidence:** 3

**Summary:**

The paper presents TeLLMate, a TEE-based protection scheme that identifies and encloses only the critical layers of an LLM inside the enclave to resist retraining-based model-stealing attacks. Concretely, TeLLMate analyzes outlier activations to locate critical layers and then runs an iterative auto layer-selection algorithm (with cosine-similarity guidance) to choose the minimal consecutive subset of layers to protect, reducing secure-memory and TEE/REE communication costs. Across LLaMA-2/3, Mistral-7B, and Qwen-3-8B, protecting small amount of parameters forces the attacker’s retrained surrogate to ≥15× higher perplexity and near random-guess zero-shot accuracy, and the defense remains robust even if the attacker initializes from public weights.

**Strengths:**

-The paper addresses an important and timely problem of defending LLMs against retraining-based model stealing under TEE constraints.
-The idea of identifying and protecting critical layers based on outlier activations is novel and intuitive.
-Experiments across multiple LLM architectures demonstrate strong defense with minimal parameter protection
-The writing is clear and the motivation and methodology are easy to follow.

**Weaknesses:**

Overall, my main concern lies in the incomplete and insufficient experimental evaluation, which fails to substantiate the authors’ central claims about efficiency and practicality.

-The experimental evaluation lacks discussion or measurement of the TEE↔REE communication overhead, even though the method is proposed to mitigate this issue.  Moreover, the experiments only focus on protection effectiveness (e.g., perplexity, accuracy) but ignore the runtime or resource implications critical to the paper’s motivation.

-No comparison is provided against existing TEE-based defenses such as TEESlice, making it unclear whether the claimed efficiency improvement actually holds.

-The analysis is shallow—there are no ablations on factors such as the number or position of protected layers, which could provide deeper understanding of the mechanism.

**Questions:**

The paper’s main motivation is to mitigate the communication overhead between the TEE and REE compared with prior systems such as TEESlice. However, the experiments mainly focus on protection effectiveness, without reporting any results related to latency, runtime, or communication cost. Could you provide additional experiments or analysis to support your claim of improved efficiency? Moreover, have you compared your method with baseline approaches like TEESlice in terms of both protection strength and efficiency? Such results would be crucial to verify that the proposed layer-wise protection indeed achieves better practicality rather than only stronger defense.

---

### Official Review · Reviewer_v765 · 2025-10-24

**Soundness:** 2
**Presentation:** 4
**Contribution:** 2
**Rating:** 2
**Confidence:** 4

**Summary:**

Trusted execution is the setup where on-device neural networks are partly executed in a slower but secure environment (TEE), which prevents leakage of sensitive data. This paper studies the placement of LLM layers between TEE environments and non-TEE environments to find the pareto optimal between resistance to model stealing attacks and latency.

Using methods from outlier activation detection, the paper proposes an algorithm to select the most critical layers, along with its neighbours to reduce the risk of model extraction. With a fixed attack budget the authors show that TeLLMate models are quite robust to model extraction, with 10-30x increases in perplexity.

**Strengths:**

1. The paper focuses on an important and timely topic: protecting LLMs deployed in edge devices from model theft.

2. The paper is generally well written and easy to follow. I appreciate the presence of motivational experiments in Section 4/5 before the main experimental section to help justify design choices.

3. The paper has an interesting technical connection between outlier activations and layer importance for model extraction attacks. Under some assumptions (more in weaknesses), the paper does good ablations to validate the choice of protected layers.

**Weaknesses:**

1. **Very large perplexity numbers may point to a bug**: Some of the perplexity numbers reported are quite absurd, and may point towards extremely unstable training more than a model extraction failures. N-gram LLMs can easily achieve ~140 perplexity on PTB (https://arxiv.org/pdf/1412.7753), and LSTMs can touch 50s (https://arxiv.org/pdf/1708.02182). The paper reports 3559.06 on PTB for some attacks, which is quite suspicious and may point towards bugs in training setup (or related to budget/model size).

2. **Why do attackers need to have a query / surrogate size budget?**: the model extraction being studied here is happening on a device the attacker has access too, unlike the typical cloud / API setting where attackers can get throttled. Given this, why should their query budget be fixed? While I agree this is useful for validating the relative strengths of different TeLLMate variants, it would be useful to see how much better the attacks get if the query budgets are significantly increased. Related question, what if a deeper network is used as the surrogate to aid learning?

3. **E2E extraction attacks maybe more effective than layer copying**: my understanding is that all attacks in the paper attempt to copy individual layers in the TEE. This may be fundamentally limiting since the layers before/after are fixed and it's a less flexible optimization problem for the model. My suspicion is this is the main cause for high perplexity (weakness 1) and model collapse. How effective are the attacks at the same budget if the extraction is done e2e rather than just the TEE layers?

4. **What's the pareto frontier between latency and security for TEE layers?** It would be helpful to know how much is the latency drop for every extra layer in the TEE and plot that against the attack accuracy to get a sort of pareto frontier. Latency was a big motivating argument for dividing layers between TEE / REE but I didn't see much experimental evidence for this.

5. **No empirical comparisons with previous layer splitting methods**: the related work section cited a bunch of previous methods such as TEESlice. However all experiments were against ablated versions of the author's setup. It would be nice to compare against the closest method in existing literature, perhaps on the latency / accuracy pareto frontier.

**Questions:**

What is the surrogate model size in the main experimental setup? Is it identical to the TEE layers?

---

### Official Review · Reviewer_55Dd · 2025-10-26

**Soundness:** 2
**Presentation:** 2
**Contribution:** 3
**Rating:** 2
**Confidence:** 3

**Summary:**

The authors propose TeLLMate, a framework designed to securely deploy LLMs on edge devices using Trusted Execution Environments (TEEs). Unlike existing methods that require extensive retraining or protect all model layers, TeLLMate selectively secures only the most critical layers, achieving high security under resource constraints. It first identifies a layer essential to model performance by observing “standard deviation shift,” and then selectively opts in adjacent layers based on “layer-wise cosine similarity” until it reaches a certain level of security or a certain amount of inclusion. Experiments show that protecting at most 10% of the model’s parameters can drastically degrade a stolen model’s performance, reducing accuracy to near-random levels.

**Strengths:**

1. The scope of the work (achieving security under the resource constraint of TEE) is well explained and is easy to follow.
2. The authors relate outliers, which are usually discussed in the model compression context, to the TEE context, which is conceptually very interesting.
3. The attacker in their threat model is inspired by LLM-Streamline (a reasonably recent model compression technique), which indicates they are aware of the relevant field beyond TEE. Their defense experiment is built not on an overly naive baseline but on a reasonable setting (as shown in Table 1).

**Weaknesses:**

A. Ablations on their proposed method are mostly missing. For example,

1. **Discussion on non-adversarial context** (Section 4.3, line324-330): The paper fully focuses on discussing the post-attack PPL, and there is no discussion about how much performance drop is achieved after removing the critical component (i.e., before the retraining attack). Adding this baseline number to Table 1 would help provide better intuition about how much drop is achieved through removing the critical component, and how much is recovered by the attacker.

2. **Justification for the std-based approach** (Section 4.3, line324-330): The authors observed that there exists a critical layer based on the Wanda pruning score (which itself is promising). However, they then turn to their unique metric (Eq. 1), claiming that it solves reliance on the calibration dataset and computational intensity. I do not see why. As far as I understand, their method also relies on activations (which should be based on some calibration samples), and I’m not sure why it reduces computation.

3. **Scalability of the termination protocol** (Section 4.3, line 348-350): The authors use a constant of PPL = 300 as the threshold to terminate the layer selection. I want the authors to justify the choice. Doesn’t this lead to choosing more layers as model size increases, because it could be more robust against dropping a layer? And can’t this be the exact reason why they end up choosing more layers for the larger model (Table 2)? I believe the number of chosen layers should rather be fewer for larger models (or at least remain the same), given the resource constraint of TEE. However, their design seems to go against it.

4. **Potential over-complication of the method** (Section 4.3, line 359-370): They mention that the std-based approach (Eq. 1), which they proposed, is not enough, and cosine similarity gives a meaningful signal on how important each layer is. Here, I wondered why it is not entirely based on cosine similarity. For example, a simpler algorithm “choosing two adjacent layers with respect to cosine similarity” should lead to the same choice of layer 0–1. The experiments are not diverse enough to justify their method.

5. **Justification for the layer-selection mechanism** (Section 5.2, general): Table 2 suggests that, although Qwen is an exception, layers 0–1 are almost always critical (e.g., layer 3 is selected for Llama2-13B, but the algorithm did not reach a satisfactory level until it removes layer 0. For the other Llama models, it always starts with layer 1, with 0 being chosen as the adjacent layer.) I think the diversity of the results is again not enough to disprove this view. I would at least like to see what happens for Qwen if layers 0–1 are selected instead of 6–7. (Table 3 does some ablation in this direction, which I think is promising, but there should be something more convincing than selecting random layers.)

6. **Further doubt about the layer 0-1 issue** (Section 5.3, Table 4): They compare the PPL with and without their metric. However, given that the selected layers are 0–1 for all of the tested models here (Llama2-7B, Llama3-8B, Mistral-7B), it cannot be concluded whether their cosine approach helped, or if it is just that choosing layer 0–1 is usually a good choice.

B. (Minor) I observe several points that could improve the presentation

1. (Section 1, line 53) The abbreviation “REE” is used without introducing its full name. It is introduced for the first time in Section 2.1.

2. (Section 4.2, line 256-261) The paragraph says Table 1 contains “MLP, Layer, and Layer+Following Layer”. However, Table 1 seems to have “Outliers, MLP, and Layer”.

3. (Section 4.3, Fig.6) The blank space of Layer Index > 32, or Tensor Value >1.0, should be removed if values can never be in this range.

4. (Section 4.3, general) The authors spend more than one page explaining the algorithm. When I read this, I wanted to quickly grasp the overview. Pseudocode should help with that. I found that the authors put it in App. A.3, which is nice, so I suggest pointing to it or including a smaller version in Section 4.3.

**Questions:**

1. (Abstract)  The authors mention that at most 10% is needed to protect the model. I think the worst case refers to the result of Llama2-13B, where 4 layers are required. This means roughly a billion parameters need to be secured. Can the authors discuss whether this is practical, considering the current resource constraints of TEE?

---

### Official Review · Reviewer_eCRC · 2025-11-04

**Soundness:** 3
**Presentation:** 3
**Contribution:** 3
**Rating:** 6
**Confidence:** 4

**Summary:**

TeLLMate addresses the challenge of protecting LLMs deployed on edge devices from model-stealing attacks using TEEs. The paper identifies that removing outlier-associated weights from just a few specific critical layers causes significant performance degradation, and proposes an algorithm that selectively places only these minimal critical layers inside the TEE while keeping the rest in the REE space. The method achieves strong protection while protecting only ~6-10% of model parameters. The approach is validated across different type of LLMs and demonstrates resistance against retraining-based model stealing attacks with minimal computational overhead.

**Strengths:**

1. Very timely work, exciting to see use of TEEs to protect LLM at the edge
2. Novel work, following other recent advantages in the area (e.g., TEESlice)
3. Methodology and evaluations look reasonable

**Weaknesses:**

1. The manuscript was not easy to read. It includes complicated sentences and some grammatical errors.
2. I believe that the paper would improve by adding some LLM relating information in the Background section, for instance mentioning the architecture of LLM networks, defining the MLP block and QKV weights that are mentioned after L267.
3. While, as I mentioned above, I do like the contributions of this work, I was also expecting to see some real-world evaluations to explore the performance of your approach with TEE-enabled edge devices. This is particularly important since you mention low communication cost in your approach. Most of the papers you cite do that, and would be really interesting to see a comparison (for instance using an Nvidia Jetson machine).
4. I didn’t understand the attacker’s thread model, particularly the assumption that the model architecture is known. Everything inside a TEE is/should not be accessible to an attacker. Also, you comment “Although this is not practical in real-world scenarios, it allows for rigorous security analysis”. Considering that this is a weaker access, why do you claim that it allows “rigorous security analysis”?
5. Minor: y axis label is missing from Figure 2

**Questions:**

1. You note that assuming knowledge of the model architecture enables “rigorous security analysis,” despite this being non-practical. Could you explain how this simplification actually contributes to a more rigorous or insightful security analysis, and what limitations or assumptions are being made in this approach?
2. Could the authors provide experimental results demonstrating the performance of their approach on a TEE-enabled edge platform (e.g., NVIDIA Jetson or similar hardware), including latency, resource consumption, and accuracy?

---

### Meta-Review · Area_Chair_yiQS · 2026-01-08

**Summary:**

This paper proposes TeLLMate, a TEE-based framework that selectively protects only critical layers of LLMs to defend against retraining based model stealing while reducing secure-memory and communication overhead. Reviewers found the problem timely and the core idea of identifying critical layers via outlier analysis interesting and promising. However, multiple reviewers raised concerns about incomplete experimental validation, unclear threat model assumptions, limited ablations, and lack of comparison to prior TEE-based methods. Questions were also raised about practicality on real TEE-enabled hardware and the plausibility of some reported results.

**Reviewer Concerns:**

Reviewers concerns were not addressed since no author response was provided.



Outstanding:

- Insufficient experimental evaluation of latency, communication overhead, and real TEE deployments

- Missing or weak comparisons with existing TEE-based defenses (e.g., TEESlice)

- Limited ablations and unclear justification for key design choices and thresholds

- Concerns about the threat model assumptions and attack realism

- Questions about scalability and practicality when protecting up to ~10% of parameters

- Potential issues or inconsistencies in reported perplexity results

**Reviewer Scores:**

Reviewer scores would likely remain the same or slightly decrease since no author response was provided.

---

### Decision · Program_Chairs · 2026-01-26

Reject